# Limited Proteolysis as a Regulator of Lymphatic Vessel Function and Architecture

**DOI:** 10.3390/ijms26157144

**Published:** 2025-07-24

**Authors:** Takuro Miyazaki

**Affiliations:** Department of Biochemistry, Showa Medical University Graduate School of Medicine, Tokyo 142-8555, Japan; taku@pharm.showa-u.ac.jp

**Keywords:** calpain, ADAMTS3, ADAM17, MT1-MMP, LYVE-1, VEGF-C

## Abstract

Recent advances have highlighted the multifaceted roles of the lymphatic vasculature in immune cell trafficking, immunomodulation, nutrient transport, and fluid homeostasis. Beyond these physiological functions, lymphatic vessels are critically involved in pathologies such as cancer metastasis and lymphedema, rendering their structural and functional regulation of major interest. Emerging evidence suggests that limited proteolysis is a key regulatory mechanism for lymphatic vascular function. In dyslipidemic conditions, dysregulated calpain activity impairs lymphatic trafficking and destabilizes regulatory T cells, partly via the limited proteolysis of mitogen-activated kinase kinase kinase 1 and inhibitor of κBα. In addition, a disintegrin and metalloprotease with thrombospondin motifs-3-mediated proteolytic activation of vascular endothelial growth factor-C has been implicated in both developmental and tumor-associated lymphangiogenesis. Proteolytic shedding of lymphatic vessel endothelial hyaluronan receptor-1 by a disintegrin and metalloprotease 17 promotes lymphangiogenesis, whereas cleavage by membrane-type 1 matrix metalloproteinase inhibits it. This review is structured around two core aspects—lymphatic inflammation and lymphangiogenesis—and highlights recent findings on how limited proteolysis regulates each of these processes. It also discusses the therapeutic potential of targeting these proteolytic machineries and currently unexplored research questions, such as how intercellular junctions of lymphatic endothelial cells are controlled.

## 1. Introduction

The lymphatic vasculature is responsible for maintaining fluid homeostasis within the body. This is achieved by the absorption of interstitial ingredients and water that have extravasated from blood capillaries, and their subsequent return to the circulatory system via lymphatic capillaries and collecting vessels, which is driven by skeletal muscle contractions. The lymphatic vasculature plays a crucial role in immune surveillance by facilitating the transport of lymphocytes and antigen-presenting dendritic cells to lymph nodes, where immune responses are initiated [1,2]. Lymphatic capillaries are solely composed of lymphatic endothelial cells in the absence of a basement membrane, surrounding pericytes, and smooth muscle cells. It was recently noted that lymphatic endothelial cells present antigens to prime lymphocytes into active states [1,3,4,5]. This function enables lymphatic endothelial cells to shape adaptive immune responses by presenting self- and foreign antigens via MHC class I and II molecules. In lymph nodes, LECs can induce peripheral tolerance by deleting or anergizing naïve CD8^+^ T cells, while also interacting with regulatory T cells to suppress autoimmunity. Conversely, in inflamed or tumor-associated contexts, LECs may modulate antigen availability and influence effector T cell activation or retention [6,7,8]. Beyond immune surveillance and homeostatic roles, lymphatic vessels are involved in tumor metastasis, lymphedema, and hypercholesterolemic inflammation [9,10,11]. Peritumoral lymphatic vessels enable tumor cell dissemination to lymph nodes, while vessel obstruction or absence causes lymphedema, a chronic disability often linked to cancer treatments [12]. As a result, lymphatic vessels play a pivotal role in the maintenance of biological functions and the pathogenesis of diseases. Increasing evidence indicates that limited proteolysis plays an important role in the regulation of lymphatic function. Consequently, this review aims to explore the regulatory mechanisms of lymphatic vascular function, with a particular focus on the most recent findings concerning the regulation of lymphatic trafficking, immunomodulation, and lymphangiogenesis by limited proteolysis.

**Figure 1 ijms-26-07144-f001:**
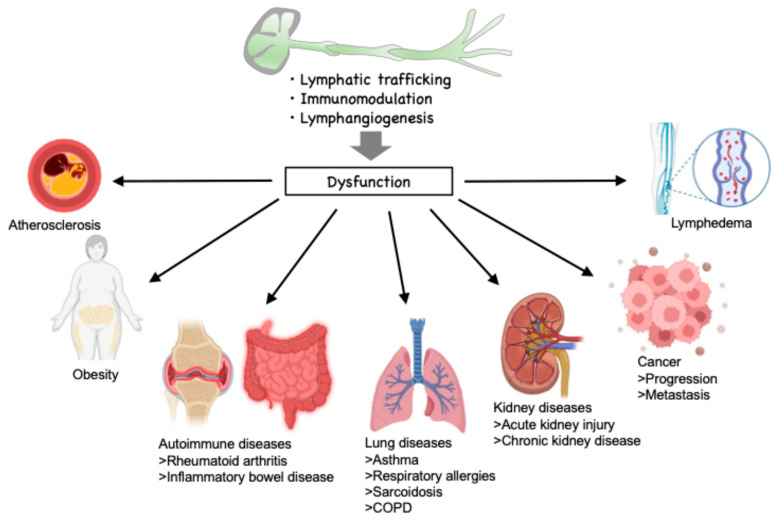
Function of lymphatic vessels and related disease. The lymphatic trafficking of immune cells significantly affects the progression of various diseases, including atherosclerosis, obesity, autoimmune diseases, cancer, lung disease, and renal disease. Lymphatic endothelial cells influence immunomodulation by immune cells, which influences the progression of atherosclerosis. Moreover, lymphangiogenesis plays a pivotal role in lymphedema and cancer metastasis. COPD: chronic obstructive pulmonary disease.

## 2. Limited Proteases in the Lymphatic Environment

Proteins within living organisms are subject to cleavage by intracellular proteases, which can be categorized into two distinct classes. The first class comprises proteases that degrade substrates to the amino acid level, including the ubiquitinolytic and lysosomal degrading systems [13,14,15,16]. The second class encompasses proteases that cleave one or multiple sites on substrates to modify them, such as caspases and calpains [17,18,19,20,21]. The former are instrumental in the control of intracellular protein quality and play a pivotal role in the regulation of amino acids and other nutrients. The latter, on the other hand, are defined as limited proteolysis and have been reported to regulate substrate function by controlling the activation, inactivation, and stability of substrates. Such intracellular proteases can be defined as limited proteases.

In addition to the intracellular proteases noted above, a wide variety of secretory proteases, including a disintegrin and metalloproteinase with thrombospondin motifs (ADAMTS) and matrix metalloproteinases (MMPs), as well as the membrane-anchored protease ADAMs, have been discussed within each protease family [22,23,24,25,26]. These ADAM proteases are known for their role in ectodomain shedding, whereby they cleave and release the extracellular domains of membrane-bound proteins. This process modulates the stability, activity, and availability of a wide range of signaling molecules, including cytokines, growth factors, and their receptors. For instance, ADAM17 (also known as TACE) regulates inflammatory responses by cleaving pro-TNF-α and activating its soluble form [27]. Through such proteolytic events, ADAM proteases critically influence intercellular communication, immune regulation, and tissue remodeling. Such extracellular proteases and ectoproteases are also classified as limited proteases.

## 3. Regulation of Lymphatic Inflammation by Limited Proteolysis

The lymphatic system plays a pivotal role in inflammation by orchestrating lymphatic trafficking of immune cells and modulating immune responses within inflamed tissues. During inflammation, lymphatic vessels actively facilitate the transport of antigen-presenting cells and lymphocytes from peripheral sites to draining lymph nodes, thereby promoting immune surveillance and adaptive immunity. Concurrently, lymphatic endothelial cells contribute to immunomodulation by producing cytokines and expressing surface molecules that influence leukocyte activation, tolerance induction, and resolution of inflammation. Dysregulation of these lymphatic functions can lead to persistent inflammation and the development of chronic inflammatory or autoimmune diseases, highlighting the lymphatic system as a crucial therapeutic target in immunoinflammatory disorders. Recently, limited proteolysis has been reported to be involved in lymphatic trafficking and immunomodulation.

### 3.1. Lymphatic Trafficking

Lymphatic vessels play a pivotal role in transporting substances and immune cells within the body. The transportation of immune cells through lymphatic vessels exerts a substantial influence on the composition of immune cells within organs, thereby regulating the progression of various diseases, including atherosclerosis, obesity, autoimmune diseases, cancer, lung disease, and renal disease (Figure 1) [28,29,30,31,32,33,34]. Growing evidence suggests that such immune cell trafficking is regulated by limited proteolysis. In our previous studies, an elevated proportion of lysophospholipids was identified within the lymphatic environment in the context of hypercholesterolemia [35]. The investigation revealed that lysophosphatidic acid, a lysophospholipid, when acting on lymphatic endothelial cells, results in the overactivation of calpain proteolytic systems. Calpain is an intracellular calcium-dependent protease and comprises 15 homologs in mammals. Among the calpain family, conventional calpains, which consist of two heterodimers, CAPN1/CAPNS1 (calpain-1) and CAPN2/CAPNS1 (calpain-2), are negatively controlled by calpastatin, an endogenous inhibitor of calpains (Figure 2). Given that lysophosphatidic acid downregulates calpastatin, conventional calpains are activated in lymphatic endothelial cells. In addition to lysophospholipids, calpain is known to be activated by growth factors and mechanical stress and is a limited protease that uses a wide variety of functional proteins as substrates, including receptors, phosphatases, intercellular adhesion molecules, and secretory proteins, to generate alternative forms of the proteins [36,37,38,39]. Furthermore, limited proteolysis by calpain facilitates recognition by ubiquitin ligase and promotes degradation by proteasomes in some instances, thereby decreasing substrate stability [40]. Accordingly, excessive activation of calpain may affect the amino acid composition in the microenvironment through this substrate sensitization [41]. In lymphatic endothelial cells, calpain activation has been demonstrated to reduce the stability of inhibitor of κBα and to increase susceptibility to NF-κB signaling [35]. As a result, calpain activation has been demonstrated to increase adhesion molecules VCAM-1, thereby decreasing the mobility of lymphocytes on lymphatic endothelial cells. Murine model experiments demonstrated that hypercholesterolemia led to a reduction in lymphatic drainage from the periphery. Consistent with the cell-based experiments, this reduction was rescued by downregulation of the lymphatic endothelial cell calpain system. Therefore, it is evident that the calpain system of lymphatic endothelial cells exerts a significant influence on the dynamics of lymphocytes within lymphatic vessels, operating through a process of limited proteolysis (Figure 3).

Sphingosine-1-phosphate is a bioactive sphingolipid metabolite that plays essential roles in vascular integrity, lymphocyte trafficking, and immune surveillance. Within the lymphatic system, the level of sphingosine-1-phosphate in lymph fluid is tightly regulated through a balance of synthesis, transport, and degradation. Sphingosine-1-phosphate is synthesized intracellularly by the phosphorylation of sphingosine via two sphingosine kinases, SphK1 and SphK2, which are differentially localized and regulated [42]. Among the cells that contribute to sphingosine-1-phosphate production in the lymphatic system, lymphatic endothelial cells are of central importance. These cells express SphK1/2 and actively secrete sphingosine-1-phosphate into the lymph using the transporter spinster homolog 2 (Spns2) [43]. The secretion of sphingosine-1-phosphate by lymphatic endothelial cells maintains a high concentration of sphingosine-1-phosphate in the lymph, creating a steep concentration gradient between the lymphoid organs (low sphingosine-1-phosphate) and the efferent lymphatic vessels (high sphingosine-1-phosphate). This gradient is essential for the egress of naive T and B cells from secondary lymphoid tissues [44]. In mouse models, deletion of Spns2 specifically in lymphatic endothelial cells leads to a significant reduction in lymph sphingosine-1-phosphate and impairs lymphocyte egress from lymph nodes [45]. On the other hand, sphingosine-1-phosphate is degraded by sphingosine-1-phosphate lyase (SPL), which irreversibly cleaves sphingosine-1-phosphate into phosphoethanolamine and hexadecenal [46], and by sphingosine-1-phosphate phosphatases (SPP1 and SPP2), which dephosphorylate sphingosine-1-phosphate back to sphingosine [47]. These enzymes are expressed in various cell types, including immune cells and stromal cells, and are particularly enriched in non-vascular tissues, contributing to the maintenance of low sphingosine-1-phosphate levels in the interstitial fluid. Despite the absence of direct evidence to substantiate the hypothesis that calpain mediates cellular regulation by sphingosine-1-phosphate in lymphatic endothelial cells, there are reports of studies that suspect a link between them. Recent studies have revealed a novel regulatory axis involving SphK1, sphingosine-1-phosphate, and calpain, which plays a pivotal role in modulating mitochondrial calcium dynamics and cellular metabolism [48]. SphK1 catalyzes the conversion of sphingosine to sphingosine-1-phosphate to influence diverse cellular functions, including calcium signaling via G-protein-coupled receptors (S1PR1–5). Notably, SphK1 overexpression enhances agonist-induced calcium release from the endoplasmic reticulum (ER), resulting in increased mitochondrial calcium uptake. A critical component of this pathway is the mitochondrial fusion protein Mitofusin 2 (MFN2), which maintains ER—mitochondria contacts at mitochondria-associated membranes. Overexpression of SphK1 leads to fragmentation of MFN2, an effect attributed to increased calpain activity. Importantly, expression of calpain-generated N- and C-terminal MFN2 fragments reproduces the mitochondrial calcium elevation observed in SphK1-overexpressing cells, suggesting that calpain-mediated MFN2 cleavage is a key mechanistic link. These findings highlight a previously unrecognized role of calpain as a downstream effector of lipid signaling via sphingosine-1-phosphate, bridging metabolic regulation and organelle communication. Elevated sphingosine-1-phosphate levels in lymphatic fluid contribute to cancer metastasis via lymphangiogenesis and immune modulation [49], exacerbate lymphatic dysfunction in lymphedema by disrupting sphingosine-1-phosphate gradients [50], promote chronic inflammation in diseases like rheumatoid arthritis and inflammatory bowel disease through enhanced leukocyte trafficking [51], and play a pathogenic role in autoimmune disorders such as multiple sclerosis, where sphingosine-1-phosphate receptor modulators have shown clinical efficacy [52]. This underscores calpain’s pivotal role in determining cell fate under conditions of altered sphingolipid metabolism in a variety of diseases. Investigating this mechanism in lymphatic endothelial cells and its roles in lymphatic trafficking would be of particular interest.

Lymphatic endothelial cells form specialized intercellular junctions that are essential for maintaining lymphatic vessel integrity, permeability, and immune cell trafficking [53]. The key junctional molecules include VE-cadherin, claudin-5/-11/-12, occludin, and junctional adhesion molecules (JAMs), which coordinate to form adherens and tight junctions. Under homeostatic conditions, these molecules maintain the barrier function and shape of lymphatic vessels. During inflammation, the integrity of junctions is dynamically regulated and often disrupted to facilitate the transmigration of immune cells. It is documented that cytokines, growth factors (e.g., VEGF-C/D), and mechanical forces can alter the expression and organization of junctional proteins. Although there is no direct evidence indicating whether intercellular junctions are processed by intracellular, secretory, or membrane-anchored proteases in lymphatic endothelial cells, investigations by us and others have demonstrated that VE-cadherin-mediated proteolysis by calpain is the rate-limiting step for vascular permeability in vascular endothelial cells [54,55,56,57]. Moreover, despite the limited number of cases reported, there is evidence indicating that limited proteolysis of the extracellular domain of VE-cadherin by ADAM10 contributes to the production of the soluble form of VE-cadherin [58]. Claudin-5, a critical component of tight junctions in lymphatic endothelial cells, is subject to limited proteolysis by MMP9 [59]. Furthermore, occludin is subject to cleavage by limited proteolysis, a process that is sensitive to GM6001, a broad-range MMP inhibitor [60]. In lymphatic endothelial cells, disruption of tight junctions has been observed in response to palmitate and TNF-α exposure [61,62]. While the involvement of limited proteolysis in the disruption of intercellular junctions in lymphatic endothelial cells remains to be elucidated, the potential for MMPs to regulate lymphatic junctions is a plausible hypothesis. This is further supported by the observed function of palmitic acid and TNF-α as drivers of MMPs via NF-κB signaling. Consequently, there is a strong probability that limited proteolysis exerts an effect on the intercellular junctions of lymphatic endothelial cells. This phenomenon warrants further investigation.

### 3.2. Immunomodulation

Lymphatic endothelial cells play dynamic and multifaceted roles in regulating adaptive immunity—not only by shaping the physical and chemical environment of lymphoid tissues but also by directly interacting with immune cells to maintain tolerance and modulate immune activation [4,5]. In this context, lymphatic endothelial cells are known to present peripheral tissue antigens to induce T cell tolerance and to produce immunomodulatory molecules sphingosine-1-phosphate, which support naïve T cell survival and effector T cell egress from lymph nodes. In arthritis, lymphatic endothelial cells influence T cell activation, and inflammation-induced autophagy in lymphatic endothelial cells degrades SphK1, reducing sphingosine-1-phosphate production [63]. Therefore, loss of lymphatic endothelial cell autophagy increases Th17 cell migration and joint inflammation, highlighting autophagy’s role in lymphatic endothelial cell-mediated immune regulation during inflammation. Recent studies have revealed that SphK1 undergoes proteolysis downstream of the tumor suppressor p53 in response to DNA damage [64]. Specifically, p53-dependent activation of the protease caspase-2 is required for SphK1 degradation, which significantly impacts endogenous sphingolipid levels. In p53-mutant triple-negative breast cancer cells, caspase-2 is not activated, and SphK1 is not degraded. However, inhibition of checkpoint kinase 1 (CHK1) in these cells restores caspase-2 activity and promotes SphK1 proteolysis, suggesting that SphK1 may be a key effector of the CHK1-suppressed apoptotic pathway. It is noteworthy that the restriction of caspase-2 activity has also been observed in other cancer types, including colorectal cancer and small cell lung carcinoma [65,66]. Therefore, further investigation into SphK1 stability beyond the context of breast cancer is warranted. Although the role of caspase signaling in sphingosine-1-phosphate production in lymphatic endothelial cells is elusive, these findings underscore the crucial role of protease-mediated SphK1 degradation in sphingolipid metabolism.

Regulatory T cells are essential for maintaining immune balance by suppressing inappropriate immune activation [67]. Recent advances in immunology highlight the regulatory roles of lymphatic endothelial cells in regulatory T cell-mediated immunomodulation [68]. Emerging evidence highlights the underappreciated role of proteases in regulating the function of regulatory T cells. Recent studies on proteases such as ADAM10, ADAM17, and mucosa-associated lymphoid tissue lymphoma translocation protein 1 reveal their significant roles in regulating regulatory T cell function [69]. These proteases may function in concert with furin, which is a proprotein convertase that cleaves and activates a variety of precursor proteins within the trans-Golgi network, endosomes, and at the cell surface [70]. It plays a critical role in processing hormones, growth factors, and viral glycoproteins, thereby regulating numerous physiological and pathological processes. These proteases are upregulated upon T cell activation and cleave pro-TGF-β1 to its active form, essential for regulatory T cell suppressive activity. Regulatory T cell-specific deletion of furin impairs this function by reducing active TGF-β1 levels [71,72]. Furthermore, metalloproteases ADAM10 and ADAM17 regulate the surface expression of lymphocyte activation gene-3 (LAG-3), a molecule related to regulatory T cell-mediated suppression [73]. Inhibiting these proteases may enhance LAG-3 expression and regulatory T cell function, particularly in mucosal immunity, as seen in colitis models [74]. In addition to regulatory T cells, it has been reported that calpain may contribute to TGF-β1 activation in vascular endothelial cells [75]. Specifically, calpain facilitates the cleavage of latent TGF-β1 complexes to convert them into their active form, leading to increased Smad2/3 phosphorylation and promotion of fibrotic remodeling associated with pulmonary hypertension. These findings underscore the therapeutic potential of targeting proteases to bolster regulatory T cell stability and immune tolerance in diseases such as autoimmune colitis and graft versus host disease.

We have previously elucidated the regulation of regulatory T cell functions centered on limited proteolysis in lymphatic vessel endothelial cells [35]. The co-culture of lymphatic endothelial cells has been demonstrated to increase the prevalence of regulatory T cells in CD4-positive T cells. This stabilization of regulatory T cells is inhibited by the knockdown of calpain in lymphatic endothelial cells using small interfering RNA (siRNA). As noted above, calpains in lymphatic endothelial cells are activated in hypercholesterolemia, thereby destabilizing IκBα. In addition to this, mitogen-activated kinase kinase kinase 1 (MEKK1) is destabilized by calpain-induced limited proteolysis. MEKK1 is a molecule upstream of JNK and ERK. As this molecular axis is also a pathway for TGF-β1 production, excessive calpain activation can decrease the production of TGF-β1 under hypercholesterolemia. As the stabilization of regulatory T cells observed in the co-culture experimental systems is inhibited by TGF-β type 1 receptor inhibitors, it is demonstrated that the TGF-β1 production pathway of lymphatic endothelial cells is involved in these phenomena. It is noteworthy that mice deficient in calpain in lymphatic endothelial cells exhibit an elevation of plasma TGF-β1 and expansion of circulating regulatory T cells. This expansion can be counteracted by the administration of TGF-β type 1 receptor inhibitors. The induction of dyslipidemia by the intercrossing of *Ldlr*-deficient mice with mice lacking calpain in lymphatic endothelial cells has been demonstrated to result in reduced atherosclerosis development in comparison with flox mice. The immunosuppressive cytokines IL-4 and IL-10, which are expressed in atherosclerotic lesions, are increased, whilst the presence of pro-inflammatory macrophages is reduced, thus suggesting increased immunosuppression by regulatory T cells in calpain-deficient mice (Figure 3). On the basis of the phenotype exhibited by the knockout mice, it can be deduced that transcriptional regulation is the predominant factor. In any case, calpain systems of lymphatic endothelial cells are hypothesized to modulate immunosuppression by regulatory T cells through the regulation of TGF-β1.

## 4. Regulation of Lymphangiogenesis by Limited Proteolysis

Lymphangiogenesis is the process of new lymphatic vessel formation, which is initiated by the proliferation of lymphatic endothelial cells and tube formation in existing capillary lymph vessels. This phenomenon is of significant importance, as it not only affects the development of the lymphatic system, but also the maintenance of capillary lymph vessels post-development. Furthermore, it plays a crucial role in the regulation and recovery of pathological conditions, such as lymphedema and cancer metastasis [76,77]. Receptor tyrosine kinase VEGFR-3 (Flt4) and its ligand VEGF-C are critical for lymphatic endothelial cell proliferation, survival, and migration. Initially expressed in blood vascular endothelial cells, VEGFR-3 becomes restricted to lymphatic endothelial cells post-midgestation [78,79,80]. VEGF-C, produced by vascular smooth muscle and mesenchymal cells, guides lymphatic endothelial sprouting [79,81]. Homozygous *Vegfc* deletion results in the absence of lymphatic vasculature and embryonic lethality, while *Vegfc*^+/−^ mice exhibit lymphatic hypoplasia [81]. *Vegfr3* deletion impairs blood vessel remodeling and causes embryonic lethality [80], while *Chy* mice with a missense mutation in VEGFR-3 display lymphatic hypoplasia and hind limb lymphedema [82]. VEGFR-3-Ig fusion protein overexpression inhibits skin lymphatic growth, whereas internal lymphatic vessels regenerate postnatally [83]. Postnatal lymphatic capillaries require VEGFR-3 activation for two weeks, after which they become VEGFR-3 independent [84]. Therefore, VEGF-C/VEGFR-3 signaling is essential for the development, maintenance, and functional regulation of the lymphatic vasculature, particularly through its critical role in lymphatic endothelial cell proliferation, migration, and survival.

Hennekam lymphangiectasia-lymphedema syndrome (Online Mendelian Inheritance in Man 235510) is a rare autosomal recessive disease, which is associated with mutations in the collagen- and calcium-binding epidermal growth factor domains 1 (CCBE1) gene. Deficiency in either *Ccbe1* or *Vegfc* exhibits phenotypical similarity in embryonic and adult lymphangiogenesis. Jeltsch et al. noted that, while CCBE1 itself does not process VEGF-C, it promotes the production of the 29/31-kDa form of VEGF-C, which is otherwise poorly active [85]. Proteolytic cleavage of VEGF-C by a disintegrin and metalloprotease with thrombospondin motifs-3 (ADAMTS3) results in the mature 21/23-kDa form of VEGF-C, which induces increased VEGF-C receptor signaling. It is likely that pro-VEGF-C binding to VEGFR-3 is assisted by the *N*-terminal domain of CCBE1. Furthermore, adenoviral vector-mediated transduction of CCBE1 into mouse skeletal muscle enhanced lymphangiogenesis induced by VEGF-C. These results reveal that ADAMTS-3 is a protease that activates VEGF-C and that CCBE1 mediates this process to regulate lymphangiogenesis (Figure 4).

ADAMTS proteins constitute a superfamily of 26 secreted molecules comprising two related yet distinct families [86]. Among the ADAMTS genes known in humans and mice, 19 encode zinc metalloproteinases, which frequently recognize extracellular matrices as their substrates. Despite the ambiguity surrounding the physiological function of ADAMTS3, previous genome-wide association studies have indicated a potential involvement of this molecule in bronchodilator response [87], lipoprotein subclasses, triglyceride measurement [88], and height [89]. The ADAMTS3 variants c.503>C (p.Leu168Pro) and c.872>C (p.Ile291Thr), as well as the CCBE1 variant, are of particular interest as causative factors in Hennekam lymphangiectasia–lymphedema syndrome. While these two types of ADAMTS3 mutants are recessive variants, they have also been observed to induce symptoms when they are compound heterozygotes. The limited proteolysis of VEGF-C observed with the introduction of wild-type ADAMTS3 was suppressed by introducing these mutants individually into HEK293 cells [90]. In contrast to the results observed in mice, the targeted deficiency of *Adamts3* in zebrafish does not induce lymphatic defects [91]. Concurrent targeting of *Adamts3* and *Adamts14* has been observed to disrupt lymphatic development, yielding an exceptionally clean phenotype in *Adamts3*/*Adamts14* double mutants, which fully phenocopies that of *Ccbe1* and *Vegfc* mutants [92,93,94]. This finding suggests that, in fish, both ADAMTS3 and ADAMTS14 possess the capacity to activate VEGF-C. The demonstration that a single wild-type copy of either gene is sufficient to trigger lymphangiogenesis suggests that both proteases act redundantly. Furthermore, zebrafish ADAMTS14 was found to be able to activate human VEGF-C, which prompted a re-examination of the VEGFC cleavage capacity of human ADAMTS14 in vitro. Notably, it is not clear whether ADAMTS14 has a redundant function with ADAMTS3 in humans. These results revealed that human VEGFC is not only proteolytically processed upon the addition of ADAMTS3, as described above [85], but also in the presence of recombinant ADAMTS14 protein in the conditioned medium. This suggests that the processing capacity of both ADAMTS proteases is an evolutionarily conserved feature. Experiments involving the transplantation of cells have demonstrated that neuronal structures and/or fibroblasts constitute cellular sources for both proteases as well as CCBE1 and VEGF-C [91]. Nevertheless, future studies will elucidate the potential involvement of ADAMTS14 in other aspects of lymphatic development. Hence, ADAMTS3 and ADAMTS14 appear to function redundantly in lymphangiogenesis through their conserved capacity to proteolytically activate VEGF-C, highlighting their evolutionary role in lymphatic development and underscoring the need for further investigation into the broader functions of ADAMTS14.

The pathophysiological importance of the proteolytic activation of VEGF-C is not fully understood, but recent studies provide causative evidence linking this process to colorectal cancer. Song et al. demonstrated that the nuclear localization of the transcription factor TEAD4 is associated with lymphatic metastasis and high lymphatic vessel density in patients with colorectal cancer [95]. Mechanistically, this is due to the formation of complexes between Yap/TAZ and TEAD4 in both colorectal cancer cells and cancer-associated fibroblasts, which directly enhance CCBE1 transcription through interaction with its enhancer regions. This upregulation of CCBE1 leads to increased proteolytic activation of VEGF-C, thereby promoting tube formation and migration of human lymphatic endothelial cells in vitro and lymphangiogenesis in a colorectal cancer cell-derived xenograft model in vivo. Furthermore, the bromodomain and extraterminal domain (BET) inhibitor JQ1 was shown to significantly suppress CCBE1 transcription, inhibit VEGF-C proteolysis, and reduce tumor lymphangiogenesis both in vitro and in vivo [96]. Collectively, these findings establish a causative link between the Yap/TAZ-TEAD4-BRD4 complex and VEGF-C activation in promoting tumor lymphangiogenesis and highlight the potential of BET inhibitors as therapeutic agents targeting this pathway.

In addition to VEGF-C, it has been established that VEGF-D is a known ligand for VEGFR3, though its function does not appear to be completely redundant. Indeed, targeted *Vegfd* deficiency does not affect lymphatic development, and VEGF-C or VEGF-D overexpression induces cutaneous lymphangiogenesis without affecting blood vessels [97,98]. In vitro studies have revealed that VEGF-D undergoes limited proteolysis following secretion, a process that has been linked to the involvement of ADAMTS3 and CCBE1. Previous investigations have demonstrated that the formation of an ADAMTS3-CCBE1 complex can occur independently of VEGFR3 and is essential for the conversion of VEGF-C, but not VEGF-D, into an active form [99]. Moreover, gene targeting studies in mice revealed that ADAMTS3 is required for lymphatic development in a manner that is identical to the requirement of VEGF-C and CCBE1 for lymphatic development. Moreover, CCBE1 was required for in vivo lymphangiogenesis stimulated by VEGF-C but not VEGF-D. Collectively, these studies elucidate that lymphangiogenesis is subject to regulation by two discrete proteolytic mechanisms of ligand activation: one in which VEGF-C activation by ADAMTS3 and CCBE1 exhibits spatial and temporal patterns in developing lymphatics, and one in which VEGF-D activation by a distinct proteolytic mechanism may be stimulated during inflammatory lymphatic growth. Thus, the proteolytic activation of VEGF-C and VEGF-D may have different physiological roles, but this point will require further investigation.

In addition to the proteolytic activation of growth factors, limited proteolysis may contribute to lymphangiogenesis. A previous study established that the ectodomain of lymphatic vessel endothelial hyaluronan receptor-1 (LYVE-1) undergoes proteolytic cleavage, resulting in the production of soluble LYVE-1 [100]. Furthermore, the cleavage site for ectodomain shedding was identified, and an uncleavable mutant of LYVE-1 was generated. In lymphatic endothelial cells, ectodomain shedding of LYVE-1 was triggered by VEGF-A. VEGF-A-induced LYVE-1 ectodomain shedding was mediated through ADAM17. Furthermore, wild-type LYVE-1, but not uncleavable LYVE-1, promoted migration of lymphatic endothelial cells in response to VEGF-A. Immunohistochemical analyses in human psoriasis lesions and dermis in VEGF-A transgenic mice suggested that the ectodomain shedding of LYVE-1 occurred in lymphatic vessels undergoing chronic inflammation. These results indicate that the ectodomain shedding of LYVE-1 might be involved in promoting pathologically induced lymphangiogenesis (Figure 4). On the other hand, limited proteolysis of LYVE-1 via MMP may have an inhibitory effect on lymphangiogenesis. Indeed, Wong et al. reported that membrane type 1-MMP (MT1-MMP) acts as an endogenous suppressor of lymphatic vessel growth [101]. MT1-MMP is synthesized from the MMP14 gene and possesses a transmembrane domain. This molecule degrades the extracellular matrix and converts pro-MMP2 into its active form. Since MT1-MMP-deficient mice exhibit defective fibroblast growth factor-2 (FGF2)-induced corneal angiogenesis [102,103] and impaired blood vessel invasion during endochondral ossification, MT1-MMP is recognized as a crucial regulator of blood vessel growth. Additionally, these mice display corneal developmental lymphangiogenesis without changing angiogenesis. Mice lacking MT1-MMP in either lymphatic endothelial cells or macrophages demonstrate corneal lymphangiogenic phenotypes similar to those observed in systemic *Mmp14*-deficient mice. This suggests that spontaneous lymphangiogenesis is autonomous to lymphatic endothelial cells and associated with macrophages. Mechanistically, MT1-MMP directly cleaves LYVE-1 on lymphatic endothelial cells, thus inhibiting LYVE-1-mediated lymphangiogenic responses. LYVE-1 is the primary receptor for hyaluronan on the surface of lymphatic endothelial cells and serves as a key specific marker for these cells [104]. Hyaluronan is the major glycosaminoglycan in the extracellular matrix and is abundantly found in connective and epithelial tissues. The interaction of hyaluronan with LYVE-1 activates intracellular signaling, promoting lymphangiogenesis in vitro [105]. Hyaluronan-induced Akt phosphorylation is inhibitable by a LYVE-1 mutation that is resistant to MT1-MMP-induced proteolysis [101]. Consequently, the limited proteolysis of LYVE-1 has the potential to impede downstream signaling after the binding of hyaluronan to its receptor. Ingvarsen et al. have developed an antibody that can inhibit the ability of the enzyme to activate pro-MMP-2 without impairing the collagenolytic function and general proteolytic activity of MT1-MMPs [106]. Using this antibody, they have demonstrated that the MT1-MMP-catalyzed activation of pro-MMP2 plays a pivotal role in the outgrowth of cultured lymphatic endothelial cells in a collagen matrix in vitro, as well as in lymphatic vessel sprouting assays conducted ex vivo. Therefore, MT1-MMP has a role as an endogenous inhibitor of developmental lymphangiogenesis and is mediated through its ability to proteolytically activate MMP2. These findings highlight the complex and context-dependent roles of limited proteolysis in regulating lymphangiogenesis. Indeed, ADAM17 and MT1-MMP function as accelerators and decelerators of lymphangiogenesis, respectively. While both proteases target LYVE-1, they have opposite roles in lymphangiogenesis. These observations suggest that site-specific proteolytic cleavage of a common substrate can elicit opposing functional outcomes, underscoring the need to consider both the identity of the protease and the physiological context in which cleavage occurs.

## 5. Therapeutic Potential and Future Perspective

Recent studies have revealed that limited proteolysis, particularly by the calcium-dependent protease calpain, plays a critical role in regulating immune cell trafficking through lymphatic endothelial cells (Figure 3). In conditions such as hypercholesterolemia, lysophospholipids like lysophosphatidic acid accumulate and downregulate calpastatin, the endogenous inhibitor of calpain, thereby promoting calpain activation [35]. These molecular changes contribute to reduced lymphocyte mobility and impaired immunosuppressive function, exacerbating inflammation and disease progression in conditions such as atherosclerosis. Importantly, genetic ablation of calpain in lymphatic endothelial cells has been shown to restore lymphatic drainage, elevate systemic TGF-β1 levels, expand regulatory T cells, and reduce pro-inflammatory macrophage infiltration in atherosclerotic lesions. This suggests that calpain activation in lymphatic endothelial cells modulates not only the lymphatic immune interface but also systemic immune tolerance via TGF-β1-mediated stabilization of regulatory T cells. These findings highlight a promising therapeutic strategy: modulating limited proteolysis by targeting calpain activity in lymphatic endothelial cells. By suppressing excessive calpain activation, it may be possible to restore TGF-β1 signaling, enhance regulatory T cell-mediated immunosuppression, and prevent the progression of chronic inflammatory diseases. Moreover, given that calpain may also activate latent TGF-β1 under certain contexts, a nuanced understanding of its context-dependent effects is essential for therapeutic development (Figure 5). Therefore, targeting limited proteolysis in lymphatic endothelial cells offers a novel and disease-modifying approach to restore immune balance in lipid-related disorders and chronic inflammation. Notably, in addition to deinhibition by lysophosphatidic acid, it appears that sphingosine-1-phosphate can also activate calpain [48]. Sphingosine-1-phosphate gradients within the lymphatic system, which are maintained by the coordinated activity of SphK1/2, Spns2, SPP1/2, and SPL, are essential for the egress and trafficking of immune cells. In light of the possible contribution of sphingosine-1-phosphate to inflammatory and autoimmune diseases, it is recommended that subsequent research prioritize the investigation of lymphatic calpain activity in these diseases. Lymphatic vessels have been demonstrated to serve as a crucial link between immune cell dynamics and the development of various diseases [28,29,30,31,32,33,34]. It is imperative to elucidate the influence of sphingolipid-mediated proteolytic signaling, particularly through calpain activation in lymphatic endothelial cells, on lymphocyte trafficking. Although there are no calpain inhibitors approved as drugs, several modalities have been investigated in clinical trials to date (Table 1). Such a strategy would be worth considering for repositioning.

The regulation of limited proteolysis presents a promising therapeutic strategy for modulating lymphangiogenesis, particularly in disorders involving lymphatic dysfunction. VEGF-C is a pivotal growth factor for lymphatic endothelial cell proliferation, survival, and migration, and its activation depends on precise proteolytic processing. From a clinical standpoint, enhancing VEGF-C activation through controlled delivery of recombinant ADAMTS3—or via gene therapy targeting CCBE1 expression—may offer novel treatments for lymphedema and lymphatic hypoplasia. Furthermore, lymphatic drainage of immune cells has been demonstrated to influence the immune cellular composition of lesions in various pathological conditions [30,31,32,33,34]. Moreover, in the context of cancer, the inhibition of this pathway has the potential to suppress pathological lymphangiogenesis in diseases such as cancer metastasis [95]. The conserved nature of this proteolytic system across species underscores its therapeutic potential and warrants further research into selective modulators of ADAMTS protease activity (Figure 5).

The regulation of lymphangiogenesis by LYVE-1 is a distinctive process. Indeed, ADAM17 functions as a facilitator, while MT1-MMP/MMP2 operates as an inhibitor. This finding provides a foundation for determining whether lymphatic vessels are in an active or stable state based on the composition of proteases in the lymph fluid. Furthermore, LYVE-1 is expressed on lymphatic vessel endothelial cells and certain myeloid cells, but not on vascular endothelial cells. It has been demonstrated that VEGF-C exerts its effects through the VEGFR-2 and VEGFR-3 receptors, thereby inducing a synergistic stimulation of angiogenesis in vascular endothelial cells [107,108]. Consequently, the targeting of LYVE-1 emerges as a more specific strategy (Figure 5). As mentioned above, preclinical studies on lymphangiogenesis are underway; however, the conduct of clinical studies is necessary for future progress in this field.

**Table 1 ijms-26-07144-t001:** Potential therapeutic strategy targeting calpain systems and their status.

Name (Modality)	Company and Institute	Target Disease	Phase	References
ABT-957 (inhibitor)	AbbVie	Alzheimer’s disease	Phase I, terminated	[109,110]
NA-184 (inhibitor)	Abliva AB	Traumatic brain injury	Phase I is being planned	[111]
AMX0114 (antisense)	Amylyx Pharmaceuticals	Amyotrophic lateral sclerosis	Phase I	[112]
SJP-0008 (inhibitor)	Senju Pharmaceutical Co., Ltd.	Central retinal artery occlusion	Phase III	[113]

## 6. Conclusive Remarks

The precise regulation of limited proteolysis can be regarded as a rational and targeted strategy for the accurate modulation of lymphangiogenic signaling in disease contexts. However, there remain areas other than hypercholesterolemia that require further investigation. For instance, inflammatory bowel disease (IBD), a chronic condition characterized by inflammation of the gastrointestinal tract, appears to be amenable to treatment through adenoviral overexpression of VEGF-C [114]. Consequently, interventions targeting lymphatic vessels have the potential to manifest in a variety of inflammatory diseases. Since limited proteolysis and lipid metabolism are closely related, the resolution of these inquiries depends on the analysis of secretory proteases in lymph fluid and lipidomic analysis. The utilization of mouse models of the aforementioned diseases, employing the Cre/loxP system (e.g., *Prox1* promoter), will help to elucidate the role of lymphatic vessels in various pathological conditions.

## Figures and Tables

**Figure 2 ijms-26-07144-f002:**
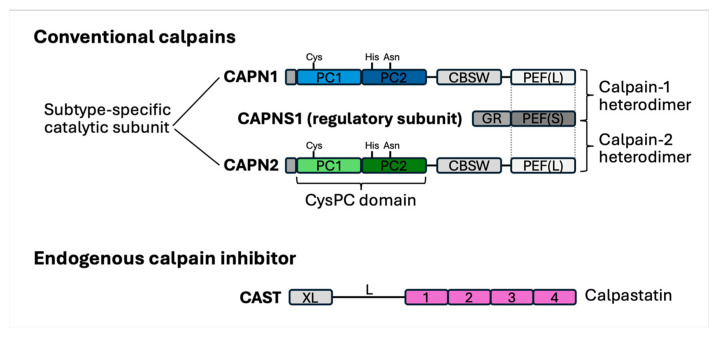
The calpain systems in lymphatic endothelial cells. Calpains represent a superfamily of cysteine proteases characterized by the presence of a CysPc domain. Two types of conventional calpain isozymes are expressed in lymphatic endothelial cells. Calpain-1 and calpain-2 are formed when the subtype-specific catalytic subunits CAPN1 and CAPN2 form heterodimers with the common regulatory subunit, CAPNS1 (calpain-s1), respectively. Conventional calpains are suppressed by calpastatin, a specific endogenous inhibitor. CBSW, calpain-type β-sandwich domain; CysPc, cysteine protease domain, calpain-type; GR, glycine-rich domain; PC, protease core; PEF, penta-EF-hand domain.

**Figure 3 ijms-26-07144-f003:**
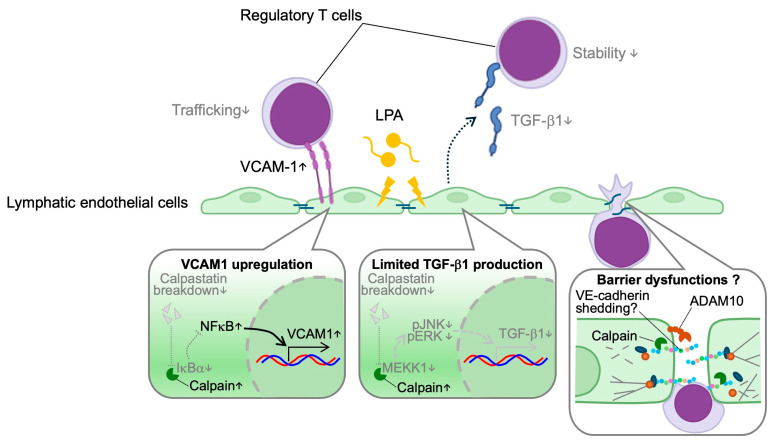
Limited proteolysis modulates lymphatic trafficking and immunomodulation. Calpain systems in lymphatic endothelial cells can be activated by lysophosphatidic acid in lymph, and proteolyses IκBα and MEKK1, resulting in upregulation of VCAM-1 and downregulation of TGF-β1, respectively. This limits the motility and the stability of lymphocytes, including regulatory T cells. It has been demonstrated that calpain and ADAM10 have the capacity to degrade VE-cadherin and diminish barrier function in vascular endothelial cells. Therefore, it can be hypothesized that these proteases may also modulate this function in lymphatic endothelial cells. ADAM10: a disintegrin and metalloproteinase 10; IκBα: inhibitor of nuclear factor-κBα; LPA: lysophosphatidic acid; MEKK1: mitogen-activated kinase kinase kinase 1; NF-κB: nuclear factor-κB; TGF-β1: transforming growth factor-β1; VCAM-1: vascular cell adhesion molecule-1; VE-cadherin: vascular endothelial-cadherin.

**Figure 4 ijms-26-07144-f004:**
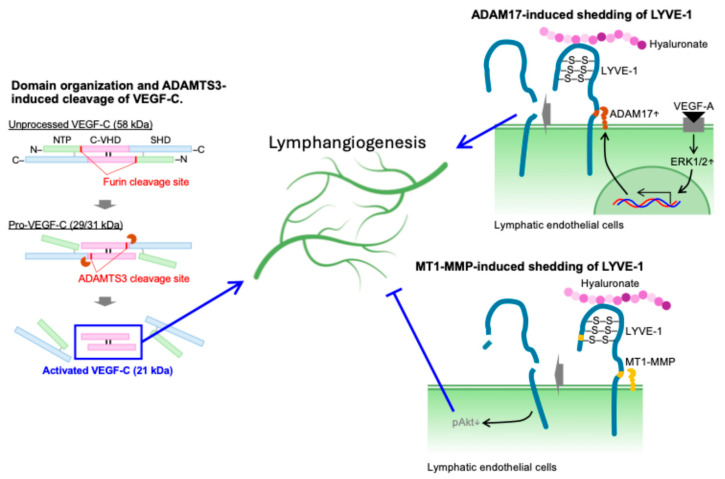
Regulation of lymphangiogenesis by limited proteolysis. ADAMTS3 promotes lymphangiogenesis via limited proteolysis of VEGF-C. Unprocessed VEGF-C is cleaved by furin and converted to pro-VEGF-C. Furthermore, pro-VEGF-C is converted to the active form via ADAMTS3-induced limited proteolysis. On the other hand, ADAM17 and MT1-MMP are involved in shedding of LYVE-1 in lymphatic endothelial cells. The former is induced by VEGF-A and promotes lymphangiogenesis. The latter suppresses lymphangiogenesis by decreasing Akt phosphorylation. In other words, limited proteolysis of LYVE-1 may induce opposite responses depending on the cleavage site. ADAM17: a disintegrin and metalloproteinase17; ADAMTS3: a disintegrin and metalloproteinase with thrombospondin motifs 3; ERK: extracellular signal-regulated kinase; LYVE-1: lymphatic vessel endothelial hyaluronan receptor-1; MT1-MMP: membrane-type 1 matrix metalloproteinase; VEGF: vascular endothelial growth factor.

**Figure 5 ijms-26-07144-f005:**
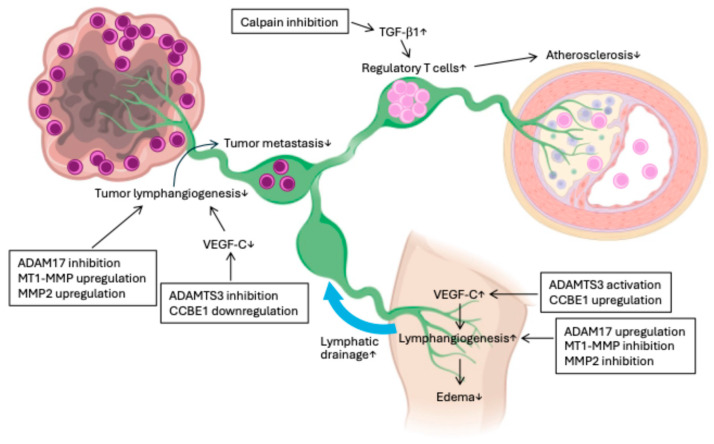
Therapeutic potential of limited proteolysis. In atherosclerosis, calpain inhibition restores TGF-β1 signaling, enhances regulatory T cells, and reduces macrophage-driven inflammation, suggesting a promising strategy for atherosclerosis treatment. In the context of cancer metastasis, ADAMTS3/CCBE1, ADAM17, and MT1-MMP/MMP2 regulate lymphangiogenesis through distinct mechanisms—MMP2 inhibits and ADAM17 facilitates LYVE-1-mediated remodeling. ADAMTS3/CCBE1 promotes lymphangiogenesis through activation of VEGF-C. Modulating these proteases can suppress tumor-associated lymphangiogenesis and metastasis. Similarly, modulating lymphangiogenesis through those limited proteases may improve lymphatic regeneration and alleviate lymphedema. ADAM17: a disintegrin and metalloproteinase17; ADAMTS3: a disintegrin and metalloproteinase with thrombospondin motifs 3; CCBE1: collagen- and calcium-binding epidermal growth factor domains 1; LYVE-1: lymphatic vessel endothelial hyaluronan receptor-1; MT1-MMP: membrane-type 1 matrix metalloproteinase; VEGF: vascular endothelial growth factor.

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
