# Peer review of "Limited Proteolysis as a Regulator of Lymphatic Vessel Function and Architecture"

_ijms, 2025, doi:10.3390/ijms26157144_

Round 1
Reviewer 1 Report
Comments and Suggestions for Authors
Thank you for sharing the manuscript titled "Limited Proteolysis as a Regulator of Lymphatic Vessel Function and Architecture. This is a well-organized and comprehensive review article that synthesizes current knowledge on the role of limited proteolysis in regulating lymphatic vessel function, immune modulation, and lymphangiogenesis. The author provides an in-depth discussion supported by recent studies and introduces mechanistic insights relevant to various disease states. The manuscript is clearly written, the figures are informative, and the references are appropriate. However, some sections would benefit from improved organization, minor clarifications, and more critical appraisal of the cited studies.
Abstract (Lines 8–24)
- Consider briefly stating how the review is structured (e.g., organized into trafficking, immunomodulation, and lymphangiogenesis). Also, clarify what “unexplored research questions” are meant—perhaps hint at one.
Introduction (Lines 27–67)
- Line 35–37: Good mention of antigen presentation by lymphatic endothelial cells. Consider expanding slightly on how this impacts adaptive immunity.
- Line 51–66: The classification of proteases is clear. However, this section is dense. Recommend breaking it into shorter paragraphs or adding subheadings (e.g., “Types of Proteases,” “Functions in Lymphatics”).
2.1 Lymphatic Trafficking (Lines 68–181)
- Lines 79–84: The description of calpain isoforms is technical. Consider simplifying for broader readership or moving more detailed molecular information to a boxed figure or table.
- Line 100–101: The conclusion linking calpain activation to lymphocyte mobility is strong. A figure summarizing this signaling cascade (Figure 2) is well-placed.
- Lines 113–160: Excellent explanation of the S1P pathway. However, some mechanistic links between SphK1, MFN2, and calpain are speculative—clearly label as such.
- Line 180–181: Suggest stating more directly what experimental data (if any) support the effect of proteolysis on junctional proteins in lymphatic ECs.
2.2 Immunomodulation (Lines 182–257)
- Line 194–200: The discussion of caspase-2/p53/SphK1 is interesting but may need clarification on how generalizable this mechanism is beyond breast cancer cells.
- Lines 204–224: Great inclusion of furin and LAG-3, but some overlap exists with prior calpain discussion. Consider condensing for flow.
2.3 Lymphangiogenesis (Lines 258–418)
- Lines 268–274: The mention of heterozygous Vegfc+/− phenotypes is excellent—showcases clinical relevance.
- Lines 280–291: Good use of Figure 3. Clarify the mechanism by which MT1-MMP inhibits Akt phosphorylation—does this come from a cited study?
- Lines 303–325: The distinction between ADAMTS3 and ADAMTS14 is well explained. Clarify if human data support ADAMTS14’s role or if limited to zebrafish.
- Lines 341–356: The link to colorectal cancer and the TEAD4–Yap/TAZ axis is compelling. However, emphasize whether these are correlative or causative findings.
- Lines 390–417: The dual role of LYVE-1 proteolysis is nicely presented. However, recommend a clearer summary comparing ADAM17 vs. MT1-MMP actions.
Section 3: Therapeutic Potential and Future Perspective (Lines 419–473)
- Include a small table summarizing potential therapeutic targets (e.g., calpain inhibitors, ADAMTS3 delivery, LYVE-1 cleavage inhibitors), their status (preclinical/clinical), and associated diseases.
Section 4: Conclusive Remarks (Lines 474–482)
- Line 475–476: Excellent idea to extend beyond hypercholesterolemia. Consider giving one example of another condition (e.g., inflammatory bowel disease).
- Line 480–482: Good suggestion about using Cre-lox models. Recommend naming specific tissue-specific Cre drivers (e.g., Prox1-CreERT2).
Author Response
[Author] Thank you so much for your valuable comments and suggestions. Here I will address your question and suggestion in a point-to-point manner.
Thank you for sharing the manuscript titled "Limited Proteolysis as a Regulator of Lymphatic Vessel Function and Architecture. This is a well-organized and comprehensive review article that synthesizes current knowledge on the role of limited proteolysis in regulating lymphatic vessel function, immune modulation, and lymphangiogenesis. The author provides an in-depth discussion supported by recent studies and introduces mechanistic insights relevant to various disease states. The manuscript is clearly written, the figures are informative, and the references are appropriate. However, some sections would benefit from improved organization, minor clarifications, and more critical appraisal of the cited studies.
Abstract (Lines 8–24)
- Consider briefly stating how the review is structured (e.g., organized into trafficking, immunomodulation, and lymphangiogenesis). Also, clarify what “unexplored research questions” are meant to be—perhaps hint at one.
[Author] According to this comment, the abstract was modified.
––––This review is structured around two core aspects—lymphatic inflammation and lymphangiogenesis—and highlights recent findings on how limited proteolysis regulates each of these processes. It also discusses the therapeutic potential of targeting these proteolytic machineries, and currently unexplored research questions, such as how intercellular junctions of lymphatic endothelial cells are controlled. (Lines 22-27)
Introduction (Lines 27–67)
- Line 35–37: Good mention of antigen presentation by lymphatic endothelial cells. Consider expanding slightly on how this impacts adaptive immunity.
[Author] According to this comment, the text and references were added.
––––This function enables lymphatic endothelial cells to shape adaptive immune responses by presenting self- and foreign antigens via MHC class I and II molecules. In lymph nodes, LECs can induce peripheral tolerance by deleting or anergizing naïve CD8⁺ T cells, while also interacting with regulatory T cells to suppress autoimmunity. Conversely, in in-flamed or tumor-associated contexts, LECs may modulate antigen availability and in-fluence effector T cell activation or retention [6-8]. (Lines 40-46)
- Cohen, J.N.; Guidi, C.J.; Tewalt, E.F.; Qiao, H.; Rouhani, S.J.; Ruddell, A.; Farr, A.G.; Tung, K.S.; Engelhard, V.H. Lymph node-resident lymphatic endothelial cells mediate peripheral tolerance via Aire-independent direct antigen presentation. J. Exp. Med. 2010, 207, 681-688.
- Kedl, R.M.; Tamburini, B.A. Antigen archiving by lymph node stroma: A novel function for the lymphatic endothelium. Eur. J. Immunol. 2015, 45, 2721-2729.
- Lucas, E.D.; Tamburini, B.A.J. Lymph Node Lymphatic Endothelial Cell Expansion and Contraction and the Programming of the Immune Response. Front. Immunol. 2019, 10, 36.
- Line 51–66: The classification of proteases is clear. However, this section is dense. Recommend breaking it into shorter paragraphs or adding subheadings (e.g., “Types of Proteases,” “Functions in Lymphatics”).
[Author] According to this comment, the section was divided, and a subheading was added.
––––2. Limited proteases in the lymphatic environment
Proteins within living organisms are subject to cleavage by intracellular proteases, which can be categorized into two distinct classes. The first class comprises proteases that degrade substrates to the amino acid level, including the ubiquitinolytic and lysosomal degrading systems [13-16]. The second class encompasses proteases that cleave one or multiple sites on substrates to modify them, such as caspases and calpains [17-21]. The former are instrumental in the control of intracellular protein quality and play a pivotal role in the regulation of amino acids and other nutrients. The latter, on the other hand, are defined as limited proteolysis and have been reported to regulate substrate function by controlling the activation, inactivation, and stability of substrates. Such intracellular proteases can be defined as limited proteases.
In addition to intracellular proteases noted above, a wide variety of secretory proteases, including a disintegrin and metalloproteinase with thrombospondin motifs (ADAMTS) and matrix metalloproteinases (MMPs), as well as the membrane-anchored protease ADAMs, have been discussed within each protease family [22-26]. These ADAM proteases are known for their role in ectodomain shedding, whereby they cleave and release the extracellular domains of membrane-bound proteins. This process modulates the stability, activity, and availability of a wide range of signaling molecules, including cytokines, growth factors, and their receptors. For instance, ADAM17 (also known as TACE) regulates inflammatory responses by cleaving pro-TNF-α and activating its soluble form [27]. Through such proteolytic events, ADAM proteases critically influence intercellular communication, immune regulation, and tissue remodeling. Such extracellular proteases and ectoproteases are also classified as limited proteases. (Lines 65-87)
2.1 Lymphatic Trafficking (Lines 68–181)
- Lines 79–84: The description of calpain isoforms is technical. Consider simplifying for broader readership or moving more detailed molecular information to a boxed figure or table.
[Author] According to this comment, Figure 2 was added to explain calpains. Text was modified as below.
––––Among the calpain family, conventional calpains, which consist of two heterodimers, CAPN1/CAPNS1 (calpain-1) and CAPN2/CAPNS1(calpain-2), are negatively controlled by calpastatin, an endogenous inhibitor of calpains (Figure 2). (Lines 123-126)
- Line 100–101: The conclusion linking calpain activation to lymphocyte mobility is strong. A figure summarizing this signaling cascade (Figure 2) is well-placed.
- Lines 113–160: Excellent explanation of the S1P pathway. However, some mechanistic links between SphK1, MFN2, and calpain are speculative—clearly label as such.
[Author] The text was incorporated to elucidate that the ensuing sentences represent the author's hypothesis.
––––Despite the absence of direct evidence to substantiate the hypothesis that calpain mediates cellular regulation by sphingosine-1-phosphate in lymphatic endothelial cells, there are reports of studies that suspect a link between them. (Lines 180-183)
- Line 180–181: Suggest stating more directly what experimental data (if any) support the effect of proteolysis on junctional proteins in lymphatic ECs.
[Author] Additional literature was cited as below.
––––In lymphatic endothelial cells, disruption of tight junctions has been observed in response to palmitate and TNF-α exposure [61,62]. While the involvement of limited proteolysis in the disruption of intercellular junctions in lymphatic endothelial cells remains to be elucidated, the potential for MMPs to regulate lymphatic junctions is a plausible hypothesis. This is further supported by the observed function of palmitic acid and TNF-α as drivers of MMPs via NF-kB signaling. (Lines 226-231)
- Tokarz, V.L.; Pereira, R.V.S.; Jaldin-Fincati, J.R.; Mylvaganam, S.; Klip, A. Junctional integrity and directional mobility of lymphatic endothelial cell monolayers are disrupted by saturated fatty acids. Mol. Biol. Cell. 2023, 34, ar28.
- Kakei, Y.; Akashi, M.; Shigeta, T.; Hasegawa, T.; Komori, T. Alteration of cell-cell junctions in cultured human lymphatic endothelial cells with inflammatory cytokine stimulation. Lymphat. Res. Biol. 2014, 12, 136-143.
2.2 Immunomodulation (Lines 182–257)
- Line 194–200: The discussion of caspase-2/p53/SphK1 is interesting but may need clarification on how generalizable this mechanism is beyond breast cancer cells.
[Author] To my knowledge, there are no reports of SphK1 degradation by calpase-2 in other cancer types, and it is unclear whether the proteolysis is generalizable among cancer types. So I have added a discussion on this point.
––––It is noteworthy that the restriction of caspase-2 activity has also been observed in other cancer types, including colorectal cancer and small cell lung carcinoma [65.66]. Therefore, further investigation into SphK1 stability beyond the context of breast cancer is warranted. (Lines 255-258)
- Gitenay, D.; Lallet-Daher, H.; Bernard, D. Caspase-2 regulates oncogene-induced senescence. Oncotarget. 2014, 5, 5845-5847.
- Muppani, N.; Nyman, U.; Joseph, B. TAp73alpha protects small cell lung carcinoma cells from caspase-2 induced mitochondrial mediated apoptotic cell death. Oncotarget. 2011, 2, 1145-1154.
- Lines 204–224: Great inclusion of furin and LAG-3, but some overlap exists with prior calpain discussion. Consider condensing for flow.
[Author] Related contexts were compiled to improve readability.
––––In addition to regulatory T cells, it has been reported that calpain may contribute to TGF-β1 activation in vascular endothelial cells [75]. Specifically, calpain facilitates the cleavage of latent TGF-β1 complexes to convert them into their active form, leading to increased Smad2/3 phosphorylation and promotion of fibrotic remodeling associated with pulmonary hypertension. (Line 278-283)
2.3 Lymphangiogenesis (Lines 258–418)
- Lines 268–274: The mention of heterozygous Vegfc+/− phenotypes is excellent—showcases clinical relevance.
- Lines 280–291: Good use of Figure 3. Clarify the mechanism by which MT1-MMP inhibits Akt phosphorylation—does this come from a cited study?
[Author] An explanation of the cited literature [101] regarding the mechanism by which MT1-MMP inhibits Akt phosphorylation was added.
––––Hyaluronan-induced Akt phosphorylation is inhibitable by a LYVE-1 mutation that is resistant to MT1-MMP-induced proteolysis [101]. Consequently, the limited proteolysis of LYVE-1 has the potential to impede downstream signaling after the binding of Hyaluronan to its receptor. (Lines 463-467)
- Lines 303–325: The distinction between ADAMTS3 and ADAMTS14 is well explained. Clarify if human data support ADAMTS14’s role or if limited to zebrafish.
[Author] The significance of ADAMTS14 in humans was added.
––––Notably, it is not clear whether ADAMTS14 has a redundant function with ADAMTS3 in humans. (Lines 384-385)
- Lines 341–356: The link to colorectal cancer and the TEAD4–Yap/TAZ axis is compelling. However, emphasize whether these are correlative or causative findings.
[Author] The relevant section was rewritten to emphasize that it is a causative factor.
––––The pathophysiological importance of the proteolytic activation of VEGF-C is not fully understood, but recent studies provide causative evidence linking this process to colorectal cancer. Song et al. demonstrated that the nuclear localization of the transcription factor TEAD4 is associated with lymphatic metastasis and high lymphatic vessel density in patients with colorectal cancer [95]. Mechanistically, this is due to the formation of complexes between Yap/TAZ and TEAD4 in both colorectal cancer cells and cancer-associated fibroblasts, which directly enhance CCBE1 transcription through interaction with its enhancer regions. This upregulation of CCBE1 leads to increased proteolytic activation of VEGF-C, thereby promoting tube formation and migration of human lymphatic endothelial cells in vitro and lymphangiogenesis in a colorectal cancer cell-derived xenograft model in vivo. Furthermore, the bromodomain and extraterminal domain (BET) inhibitor JQ1 was shown to significantly suppress CCBE1 transcription, inhibit VEGF-C proteolysis, and reduce tumor lymphangiogenesis both in vitro and in vivo [96]. Collectively, these findings establish a causative link between the Yap/TAZ–TEAD4–BRD4 complex and VEGF-C activation in promoting tumor lymphangiogenesis, and highlight the potential of BET inhibitors as therapeutic agents targeting this pathway. (Lines 397-412)
- Lines 390–417: The dual role of LYVE-1 proteolysis is nicely presented. However, recommend a clearer summary comparing ADAM17 vs. MT1-MMP actions.
[Author] A comparison of ADAM17 versus MT1-MMP was added in the summary section.
––––These findings highlight the complex and context-dependent roles of limited proteolysis in regulating lymphangiogenesis. Indeed, ADAM17 and MT1-MMP function as accelerators and decelerators of lymphangiogenesis, respectively. While both proteases target LYVE-1, they have opposite roles in lymphangiogenesis. These observations suggest that site-specific proteolytic cleavage of a common substrate can elicit opposing functional outcomes, underscoring the need to consider both the identity of the protease and the physiological context in which cleavage occurs. (Lines 474-480)
Section 3: Therapeutic Potential and Future Perspective (Lines 419–473)
- Include a small table summarizing potential therapeutic targets (e.g., calpain inhibitors, ADAMTS3 delivery, LYVE-1 cleavage inhibitors), their status (preclinical/clinical), and associated diseases.
[Author] As far as I could find, I could not find any clinical studies on ADAMTS3 or LYVE-1 cleavage. Texts and the table summarizing calpain inhibitors were added.
––––Although there are no calpain inhibitors approved as drugs, several modalities have been investigated in clinical trials to date (Table 1). Such a strategy would be worth considering for repositioning. (Line 514-516)
––––As mentioned above, preclinical studies on lymphangiogenesis are underway; however, the conduct of clinical studies is necessary for future progress in this field. (Line 538-540)
Table 1. Potential therapeutic strategy targeting calpain systems and their status.
Name (modality) |
Company and institute |
Target disease |
Phase |
References |
ABT-957 (inhibitor) |
AbbVie |
Alzheimer’s disease |
Phase I, terminated |
110,111 |
NA-184 (inhibitor) |
Abliva AB |
Traumatic brain injury |
Phase I, being planned |
112 |
AMX0114 (antisense) |
Amylyx Pharmaceuticals |
Amyotrophic lateral sclerosis |
Phase I |
113 |
SJP-0008 (inhibitor) |
Senju Pharmaceutical Co., Ltd. |
Central retinal artery occlusion |
Phase III |
114 |
Section 4: Conclusive Remarks (Lines 474–482)
- Line 475–476: Excellent idea to extend beyond hypercholesterolemia. Consider giving one example of another condition (e.g., inflammatory bowel disease).
[Author] A description of inflammatory bowel disease and the extensibility of lymphatic vascular therapy was added.
––––For instance, inflammatory bowel disease (IBD), a chronic condition characterized by inflammation of the gastrointestinal tract, appears to be amenable to treatment through adenoviral overexpression of VEGF-C [109]. Consequently, interventions targeting lymphatic vessels have the potential to manifest in a variety of inflammatory diseases. (Line 555-559)
- Line 480–482: Good suggestion about using Cre-lox models. Recommend naming specific tissue-specific Cre drivers (e.g., Prox1-CreERT2).
[Author] The promoter name was listed in the relevant section.
––––The utilization of mouse models of the aforementioned diseases, employing the Cre/loxP system (eg. Prox1 promoter), will help elucidate the role of lymphatic vessels in various pathological conditions. (Line 569-571)
Reviewer 2 Report
Comments and Suggestions for Authors
This manuscript, titled “Limited Proteolysis as a Regulator of Lymphatic Vessel Function and Architecture” provides a comprehensive overview of how lymphatic trafficking, immunomodulation, and lymphangiogenesis are regulated through limited proteolysis. Overall, it is well-written and informative. However, I have a few suggestions that could improve clarity and enhance the manuscript’s impact.
Comments for the Authors:
Since the manuscript discusses inflammation in various contexts, it would be beneficial to include a separate section on the role of lymphatics in inflammation. This would provide a stronger foundational understanding for readers.
In Figure 3, the title displays, "Structure and ADAMTS3-induced cleavage of VEGF-C." However, no actual protein structure is shown. I recommend revising the title to:
"Domain organization and ADAMTS3-induced cleavage of VEGF-C."
In the section titled "Therapeutic potential and future perspective," the authors could enhance the discussion by incorporating a visual summary and bringing therapeutic strategies targeting proteolytic pathways in lymphatic biology.- Please find the example below.

Author Response
[Author] Thank you so much for your valuable comments and suggestions. Here I will address your question and suggestion in a point-to-point manner.
This manuscript, titled “Limited Proteolysis as a Regulator of Lymphatic Vessel Function and Architecture” provides a comprehensive overview of how lymphatic trafficking, immunomodulation, and lymphangiogenesis are regulated through limited proteolysis. Overall, it is well-written and informative. However, I have a few suggestions that could improve clarity and enhance the manuscript’s impact.
Comments for the Authors:
- Since the manuscript discusses inflammation in various contexts, it would be beneficial to include a separate section on the role of lymphatics in inflammation. This would provide a stronger foundational understanding for readers.
[Author] The sentence was restructured to emphasize Inflammation and further defined the relationship between lymphatic vessels and inflammation.
––––3. Regulation of lymphatic inflammation by limited proteolysis
The lymphatic system plays a pivotal role in inflammation by orchestrating lym-phatic trafficking of immune cells and modulating immune responses within inflamed tissues. During inflammation, lymphatic vessels actively facilitate the transport of anti-gen-presenting cells and lymphocytes from peripheral sites to draining lymph nodes, thereby promoting immune surveillance and adaptive immunity. Concurrently, lym-phatic endothelial cells contribute to immunomodulation by producing cytokines and expressing surface molecules that influence leukocyte activation, tolerance induction, and resolution of inflammation. Dysregulation of these lymphatic functions can lead to per-sistent inflammation and the development of chronic inflammatory or autoimmune diseases, highlighting the lymphatic system as a crucial therapeutic target in im-munoinflammatory disorders. Recently, limited proteolysis has been reported to be involved in lymphatic trafficking and immunomodulation. (Lines 89-101)
- In Figure 3, the title displays, "Structure and ADAMTS3-induced cleavage of VEGF-C."However, no actual protein structure is shown. I recommend revising the title to:
"Domain organization and ADAMTS3-induced cleavage of VEGF-C."
[Author] The title in the figure was modified according to the suggestion.
- In the section titled "Therapeutic potential and future perspective,"the authors could enhance the discussion by incorporating a visual summary and bringing therapeutic strategies targeting proteolytic pathways in lymphatic biology.
[Author] A new figure (Figure 5) entitled “Therapeutic potential of limited proteolysis” was added.
Round 2
Reviewer 1 Report
Comments and Suggestions for Authors
I am satisfied with the authors’ responses to my previous comments and appreciate the revisions they have made to address the concerns raised. The manuscript is now significantly improved, and I believe it meets the standards for publication. I have no further concerns, and I recommend the paper for acceptance in its current form.